# Evaluation of the Effects of Pre-Slaughter High-Frequency Electrical Stunning Current Intensities on Lipid Oxidative Stability and Antioxidant Capacity in the Liver of Yangzhou Goose (*Anser cygnoides domesticus*)

**DOI:** 10.3390/ani10020311

**Published:** 2020-02-17

**Authors:** Xin Zhang, Morgan B. Farnell, Qian Lu, Xiaoyi Zhou, Yuhua Z. Farnell, Haiming Yang, Xiaoli Wan, Lei Xu, Zhiyue Wang

**Affiliations:** 1College of Animal Science and Technology, Yangzhou University, Yangzhou 225009, Jiangsu, China; ZFF9962@163.com (X.Z.); lMonicaq@163.com (Q.L.); zxy111111www@163.com (X.Z.); yhmdip@163.com (H.Y.); wanxl1021@126.com (X.W.); dkwzy@263.net (Z.W.); 2Joint International Research Laboratory of Agriculture and Agri-Product Safety of Ministry of Education of China, Yangzhou University, Yangzhou 225009, Jiangsu, China; 3Department of Poultry Science, Texas A & M AgriLife Research and Extension, College Station, TX 77843, USA; mfarnell@tamu.edu (M.B.F.); yfarnell@tamu.edu (Y.Z.F.);

**Keywords:** electrical stunning, meat goose, liver, color, lipid peroxidation, antioxidant capacity, SOD, CAT

## Abstract

**Simple Summary:**

Livers from meat geese are popular in China for different kinds of dishes. High-frequency electrical stunning (ES) was suggested to improve chicken meat quality in previous studies. Limited research has been performed on the effect of ES on liver quality of the meat goose. This study was performed to evaluate the effects of different current intensities on lipid oxidative stability and antioxidant capacity in the liver of Yangzhou goose (a typical meat goose) based on a typical high-frequency level (500 Hz). Stunning each goose with the current intensity at 60 V/40 mA improved lipid oxidative stability and antioxidant capacity from day 0 to day 4 and the lightness at day 0 in the liver of meat geese during cold storage. A combination of 60 V/40 mA/ 500 Hz/ 10 s per bird could be applied in the ES of Yangzhou geese in order to improve the lipid oxidative stability and antioxidant capacity in the livers.

**Abstract:**

Limited research has been performed to evaluate the effects of high-frequency electrical stunning (ES) methods on the lipid oxidative stability of the meat goose livers. This study was conducted to evaluate the effects of high-frequency-ES current intensities on lipid oxidative stability and antioxidant capacity in the liver of Yangzhou goose (*Anser cygnoides domesticus*). Forty 92-day-old male Yangzhou geese were randomly divided into five treatments (*n* = 8). Geese were not stunned (control) or exposed to ES for 10 s with alternating current (AC) at 500 Hz in a water bath. Current intensities were set at 30 V/20 mA (E30V), 60 V/40 mA (E60V), 90 V/70 mA (E90V), or 120 V/100 mA (E120V), respectively. The malondialdehyde level at day 0 was the highest in 120 V (*p* < 0.05). Antioxidant enzymes’ activity on day 2 was the highest in E60V. The 1, 1-diphenyl-2-picrylhydrazyl free radical (DPPH·) elimination ability was lower in the E120V than that in the E60V at two days and four days postmortem (*p* < 0.05). A combination of 60 V/40 mA/ 500 Hz/ 10 s per bird could be applied in the ES of Yangzhou geese to improve the lipid oxidative stability and antioxidant capacity in the livers.

## 1. Introduction

Goose is an important poultry food resource in China. The Yangzhou goose is a typical meat goose that refers to a category of goose species that are raised for the primary purpose of meat production. According to the latest survey, the number of meat goose in China was predicted to exceed 500 million in 2019 [1], producing approximately 1000 tons of liver from meat geese (meat goose liver) every year. Meat goose liver is popular in China in a variety of dishes (e.g., stewed livers, fried livers, etc.). The meat goose liver is different from foie gras (also called fatty liver), which is produced mainly by overfeeding of Landes geese [2]. The meat goose liver is cheaper than foie gras; therefore, ordinary people in developing countries like China can afford it. Moreover, its production process does not involve overfeeding, therefore being acceptable on animal welfare grounds.

However, poultry liver can be suffered from lipid oxidation as reported in Landes geese [3], chickens, and ducks [4]. Lipid oxidation refers to the reactive oxygen species, which exceeds the ability of a system to neutralize and eliminate them [5]. Lipid oxidation can lead to unpleasant odors or tastes, color deteriorations, degradation in proteins, and decline in the shelf life of food [6,7]. The factors that affect lipid oxidation in poultry products include poultry diet, oxidative stresses in the live birds, pre-slaughter stunning methods, food processing methods, and storage conditions [8,9,10,11,12]. The storage time is also an important factor for lipid oxidation. In our previous study, the malondialdehyde (MDA) level in raw thigh meat that stored at 4 °C dropped to the minimum level from day 0 (d 0) to day 6 (d 6) and recovered to the level of d 0 again at day 9 (d 9) [12].

Research shows that different pre-slaughter electrical stunning (ES) parameters affect bleed-out efficiency and lipid oxidation in poultry products [13]. In China, approximately 74% of broiler slaughterhouses adopt electrical water bath stunning before birds are slaughtered to reduce the pain of poultry, and improve product quality [14]. However, no study has been performed to study the effects of ES parameters on lipid oxidation and antioxidant capacity of meat goose liver. Previous studies have explored the effects of ES parameters on the presence of engorged blood vessels in fatty livers [15,16], and on blood loss and meat downgrading in over-fed geese [15]. Generally, a combination of high electrical frequency (350 Hz) with relatively high current intensities (80 A–85 A) can decrease the loss of liver weight due to a lower removal of engorged blood vessels in the fatty liver of geese [16]. Our previous study indicated that high frequency stunning (400 Hz–1000 Hz) could improve meat quality in chickens [17], and combinations of high ES frequency and different current intensities can affect the lipid oxidation in broiler breast meat [10]. We also observed that the ES with the combination of 60 V/40 mA/500 Hz could relieve lipid oxidation in the muscle of geese compared with geese slaughtered without stunning at day 2 (d 2) (data unpublished). Thus, we hypothesize that ES current intensity at optimal levels can reduce lipid oxidation and improve antioxidant capacity in the liver of meat geese that exposed to a typical high frequency (500 Hz).

The present study was aimed to evaluate the effects of high-frequency-ES of different electrical current intensities on lipid oxidative stability and antioxidant capacity during the cold storage from 0 day to 4 days, and the color in meat goose livers at 0 days postmortem.

## 2. Materials and Methods

### 2.1. Birds and Management

A total of 100 healthy male Yangzhou geese, 28 days old, with similar body weight, were obtained from a breeding farm. The geese were raised on the plastic slat floor of the same environment on the poultry farm of Yangzhou University. They had *ad libitum* access to feed and water. The room temperature was approximately 22 ± 5 °C, and no heat was provided. The geese were under natural daylight. Geese were fed with a corn-soybean diet with the ingredients and nutrient levels shown in Table 1. At the age of 92 days, geese were weighed, and 40 geese with the closest average body weight (3.80 ± 0.15 kg) were selected to be slaughtered according to the following procedures. All geese handling protocols used in the study were approved by the Animal Care and Use Committee of the Yangzhou University (ethical protocol code: YZUDWSY 2017-09-06). 

### 2.2. Electrical Stunning System

A set of ES line, which was equipped with current and frequency transformers (SQ series frequency variable ES machine, Chang Xun Machinery, Nanjing, China), was connected to the household current (sinusoidal alternating, AC, 220V, 50Hz) for the stunning of geese. The frequency was set at 500 Hz for each ES treatment. Voltages and current intensities were adjusted from this transformer. One electrode was connected to a steel bar, and the other was connected to a mesh plate electrode which was placed into a water bathing flume (1% sodium chloride). A multiple-birds stunning system would form a parallel circuit that would result in a share of the total electrical current intensity by all the birds in the water bath, leading to the difficulty to record the current intensity that flows through each bird. Therefore, only one goose was stunned each time in our study to ensure an accurate record of the electrical current. This circuit allowed the current to flow through the feet, heart, and head of each goose. When the goose’s head was pushed in the water bath with a long “Y” shape insulate wooden bar, a closed circuit will be formed with electricity flowing through the body of the goose. Since a loose contact between the steel bar and the shackle would affect the conductivity and electrical current, the electrical bar and the shackle were fixed together with an insulated rope to maintain good electrical conduction. The automatic stunning line was replaced by a manually operated line. Once finishing a stunning procedure, the goose’s head and neck were pushed out of the water bath with the “Y” shape insulate wooden bar.

### 2.3. Experimental Design

Before stunning, the geese were randomly divided into five groups, with eight replicates per group and one goose per replicate. Geese were sacrificed without stunning (the control) or slaughtered after ES for 10 s with alternating current (AC) at 500 Hz in a water bath which combined with 30 V/20 mA (E30V), 60 V/40 mA (E60V), 90 V/70 mA (E90V), or 120 V/100 mA (E120V). Only one goose was stunned in the stunning line for each time. The electricity was shut off immediately after stunning, and the bird was removed and slaughtered via bleeding.

### 2.4. Slaughter and Sampling

All the geese were weighed individually after fasting for eight hours, and body weights were recorded as W1 before sacrifice. Exsanguinations were performed without stunning (control) or immediately after stunning (other treatments) via severing the jugular trachea, esophagus, vein, and carotid artery on both sides of the neck. Bleeding was allowed for 5 min. After bleeding, geese were weighed again, and the body weights were recorded as W2. Blood loss = (W1 − W2)/W1 × 100%.

Three pieces of the liver (approximately 10 g each) were taken from the right side of the whole liver. One piece of the sample was immediately stored at −20 ± 1 °C (d 0), and the other two pieces were respectively stored at 4 ± 0.5 °C for two days (48 h) and four days (96 h), and then transferred to −20 ± 1 °C for analysis of lipid oxidation and antioxidant capacity. All the indicators were measured within three months. A piece of liver (approximately 10 g) was collected at the upper left side of the liver at 15 min postmortem for the determination of colors.

### 2.5. Lipid Oxidation and Antioxidant Capacity

A total of 0.50 g liver was mixed with 4.50 mL saline, homogenized in an ice water bath, and centrifuged at 4 °C with 2500 r/min for 10 min. The supernatant (10% stock solution of the liver) was stored at −20 ± 1 °C for further analysis. The concentrations of MDA and protein, antioxidant capacities of total superoxide dismutase (T-SOD), catalase (CAT), and glutathione peroxidase (GSH-PX) in the livers of geese were analyzed respectively using an thiobarbituric acid (TBA) method (MDA) assay kit, A003-1 protein kit (Coomassie brilliant blue method, A045-2), SOD kit (Hydroxylamine method, A001-3), CAT kit (Visible light method, A007-1), and GSH-PX kit (Colorimetric method, A005) according to the instructions of the kits (Nanjing Jiancheng Bioengineering Institute, Nanjing, China) using the 10% stock solution of liver. The results were calculated on the basis of the protein content in liver homogenates.

The 1, 1-diphenyl-2-picrylhydrazyl (DPPH) was purchased from Tokyo Chemical Industry Co., Ltd. (Tokyo, Japan). A DPPH free radical (DPPH·) elimination ability (DPPH·EA) was determined according to the method described by Favre et al. [18], with the following modifications. The 10% stock solution of the liver was further diluted with saline to form a 2.5% working solution of the liver. This 2.5% solution was then vortexed evenly with a 0.1 mmol/L DPPH· ethanol solution at a volume ratio of 1:10. The mixture was kept still at room temperature (25 ± 3 °C) in the darkness for 30 min and then centrifuged at 3500 r/min for 10 min. The absorbance of the supernatant was then determined at 517 nm using a microplate reader (Multiskan FC, Thermo, Waltham, MA, USA). The results were expressed as μmol of Trolox equivalent (TE) per gram of protein in liver homogenates (μmol TE/g protein).

### 2.6. Liver Color

The CIE Lab values of lightness (L*), redness (a*), and yellowness (b*) were measured in triplicates using the instrument of Chroma Meter CR-400 (Konica Minolta, Chiyoda-ku, Tokyo, Japan). Calibration were done with the standard white porcelain board (L* = 97.53, a* = 0.13, b* = 1.43). The final value of each sample is the arithmetic mean of the three points.

### 2.7. Statistical Analysis

Results are represented as “means ± standard error of the mean (SEM)”. All the data were analyzed using the one-way Analysis of Variance (ANOVA) procedure of SPSS (Ver. 20.0 for Windows, SPSS, Inc., Chicago, IL, USA). Significant differences among the treatments were determined at *p* < 0.05 by Duncan’s multiple range test. The “*p* < 0.01” indicates highly significant.

## 3. Results and Discussion

### 3.1. Lipid Oxidative Stability

Lipid oxidation is an important factor affecting the sensory and nutritional value, safety, and shelf life of animal products [19]. The MDA level is an important and significant biomarker of oxidative stress and lipid oxidation [7]. The effects of stunning methods on lipid oxidation in the liver of geese are shown in Table 2. In the present study, the MDA level of liver was higher at 0 days postmortem in treatment E120V than that in other electrical treatments (*p* < 0.01). The MDA levels in geese livers were not different among all groups on day 2 and day 4 (d 4) (*p* > 0.05). Limited research has been conducted on the ES effects on lipid oxidation of meat goose liver. Different from geese, previous studies of our team demonstrated that the MDA level in breast meat of broilers was not affected by ES methods [10] or controlled atmosphere stunning (CAS) methods at day 0 (d 0) [20]. Whereas when the refrigeration time was prolonged, MDA levels at day 1 (d 1) and day 3 (d 3) were increased by low-current and high-frequency combination compared to high-current and low-frequency combination [10], or by CAS with 40% CO_2_ compared to 79% CO_2_ at d 3 [20]. These differences among studies may be due to different ES variables, body resistance (geese vs. broilers), and measured tissues (liver vs. muscle).

The MDA level in the groups of control, E90V, and E120V all decreased at d 4 as compared with d 0 (*p* < 0.05) in the present study (Table 2). This is consistent with the changes of MDA level in breast meat at d 1 and d 9 [10] and in thigh meat at d 6 as compared with d 0 [20] in electrically stunned broilers. The decrease of MDA concentration may be due to further metabolisms of MDA through enzymatic and chemical reactions, resulting in the products of thromboxane A 2, acetaldehyde, acetate, or Schiff-base adducts as time went on [21,22]. The results of the present study suggested that lipid oxidation in the liver of geese was the highest in the E120V group at 0 days postmortem. Therefore, stunning geese with a high current intensity of 120 V (100 mA) is not recommended if the liver is to be consumed on the first day-postmortem.

### 3.2. Antioxidant Capacity Stability

The effect of ES methods on antioxidant capacity is shown in Figure 1. The DPPH· EA and the GSH-PX, CAT and T-SOD enzyme activities of each treatment group at 0 day postmortem were not affected by the treatments (*p* > 0.05, Figure 1A). However, at two days postmortem, the DPPH·EA in the E30V and E60V group in geese liver was higher than that in E120V groups (*p* < 0.05, Figure 1B). The enzyme activity of CAT at d 2 in E120V group were lower (*p* < 0.05) than that in E60V group; moreover, CAT in control was lower than that in the E30V, E60V, and E90V groups (*p* < 0.05, Figure 1B). The enzyme activity values of GSH-PX in the control, E30V and E120V groups were lower compared to that from E60V at d 2 (*p* < 0.05, Figure 1B). The enzyme activity of T-SOD in E60V was higher compared to those from other groups at d 2 (*p* < 0.05, Figure 1B). The GSH-PX, CAT, and SOD are main antioxidant enzymes that play important roles in the balance of oxidation-reduction [23]. No research has been done to study the ES effects on antioxidant capacity in meat goose liver. The activity of glutathione S-transferase (an antioxidant related enzyme) was decreased at d 3 in breast meat [10] and increased at d 1 in thigh meat of broilers that were stunned with 150 V/130 mA/60 Hz as compared with 65 V/86 mA/1000 Hz in our previous study [12]. At 4 d postmortem, the DPPH·EA in control, E30V, and E60V groups were higher than that in the E120V group (*p* < 0.05, Figure 1C). The activity of T-SOD in E120V was lower than that in other groups at d 4 (*p* < 0.05, Figure 1C). However, the activity of T-SOD was not affected by ES methods in the previous study on breast meat in broilers from d 0 to d 1 [10].

The effect of storage (4 °C) times on antioxidant capacity stability in livers of meat geese treated with different ES methods is shown in Figure 2. The enzyme activity of CAT was lower at d 2 than at d 0 in all of the treatments (*p* < 0.05), except for the treatments of E30V and E60V (*p* > 0.05). The enzyme activity of T-SOD decreased from d 0 to d 4 in almost all the treatment (*p* < 0.05) except that E60V had an equal level between d 0 and d 2. The reduction of the antioxidant enzyme may due to the proteins’ cross-linking during storage [24]. Differently, the activity of T-SOD was observed to increase at d 1 compared to d 0 in the thigh muscle of broilers that were exposed to ES [12]. The enzyme activity of GSH-PX decreased on d 2 and then rose on d 4 in all the treatments (*p* < 0.05), except that E60V had a stable level from d 0 to d 4 (*p* > 0.05). However, in the thigh muscle of broilers, the activity of glutathione S-transferase was observed to decrease at d 3 compared with d 0 [12]. These antioxidant enzymes worked together leading to a higher DPPH·EA at d 2-d 4 than d 0 in the treatment E60V (*p* < 0.05), whereas lower DPPH·EA at d 2 than d 0 and d 4 in E120V group (*p* < 0.05), indicating a higher overall antioxidant stability in E60V. Our data suggest that the 60 V/40 mA/500 Hz (for each goose) alleviated lipid oxidation and enhanced antioxidant capacity stability in goose livers as compared with traditional slaughter (no stunning) and high-current ES (120 V/100 mA/500 Hz) during cold storage from d 0 to d 4.

### 3.3. Relationship between Liver Color and Bleed-out Efficiency or Lipid Oxidation

Food color is an important criterion for the judgment of food quality by consumers. The effect of stunning methods on liver color is given in Table 3. The lightness in E30V, E60V, and E120V were higher than that in control groups (*p* > 0.05). The lightness and yellowness were, respectively, increased by 13.61% (*p* < 0.01) and 24.83% (*p* = 0.07) in the liver of E60V as compared with that in the control group at d 0. The lightness and yellowness were, respectively, 7.81% (*p* < 0.05) and 22.80% higher (*p* = 0.07) in the liver of E120V than that in the control group at d 0. Previously, liver colors were not affected by ES as compared with CAS in fatty livers from both force-fed geese [25,26] and force-fed ducks [26]. A decrease of the lightness in the fatty liver was observed in overfed geese stunned with 5 s compared with 15 s (decreased by 0.18% and 0.24%, respectively) [15]. A lower current intensity can decrease the incidence of petechial hemorrhages on the fatty liver [15]. A high electrical frequency (1200 Hz) can cause a decrease in the pink/red coloration of fatty liver lobe tips in the ganders [15]. In the present study, the combination of a high frequency (500 Hz) with moderate current intensity (60 V/40 mA for each goose) resulted in a higher lightness in liver at d 0, which may look fresher and may be more acceptable for consumers.

As shown in Table 3, bleed-out efficiency was not significantly different among the treatment groups (*p* > 0.05), consistent with the study of Gregory and Wilkins [27]. Blood residue is one of the main factors affecting the color of meat products [28] and engorgement of blood vessels in the fatty liver of geese [16]. However, since blood residue was not significantly affected by ES methods, it was not likely a direct factor to affect liver color in meat geese in the present study.

Heme is an important part of myoglobin and hemoglobin. Lipid oxidation promotes oxidation by capturing Fe^2+^ in heme, which is disadvantageous to meat color stability [29]. Lipid oxidation could make meat color darker and yellowness lower [30]. In the present study, the increase of lightness and yellowness in E120V was consistent with a higher MDA level at d 0. However, since the redness was not affected by ES methods from d 0 to d 4, whether the color changes were affected by lipid oxidation needs further study.

## 4. Conclusions

Stunning each goose with a current intensity at 60 V/40 mA/500 Hz improved the antioxidant capacity stability, lipid oxidative stability, and the lightness in the liver of meat geese during cold storage from d 0 to d 4. A stunning method with 60 V/40 mA/500 Hz for each goose can be used to improve the antioxidant capacity and reduce the lipid oxidation of liver in the pre-slaughter process of geese. Future studies are necessary to optimize ES parameters for geese with the evaluations on the effects of pre-slaughter stunning stress in meat geese. All authors have read and agreed to the published version of the manuscript.

## Figures and Tables

**Figure 1 animals-10-00311-f001:**
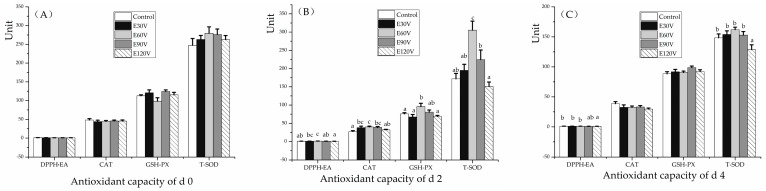
Effect of electrical stunning (ES) methods on the antioxidant capacity of meat goose livers at 0 to 4 days postmortem. Figures (**A**–**C**) respectively represent the antioxidant capacity of livers that were stored at 4 °C for 0, 2, and 4 days. Geese were slaughtered without stunning (the control) or after ES for 10 s with alternating current (AC) at 500 Hz in a water bath which combined either with 30 V/20 mA (E30V), 60 V/40 mA (E60V), 90 V/70 mA (E90V), or with 120 V/100 mA (E120V). Abbreviations: DPPH·EA: the eliminating ability of 1, 1-diphenyl-2-picrylhydrazyl free radical; CAT: catalase; GSH-PX: glutathione peroxidase; T-SOD: total superoxide dismutase. The d 0: goose liver was immediately stored at −20 ± 1 °C after slaughter; d 2 and d 4: goose livers stored at 4 ± 0.5 °C for two days (48 h) and four days (96 h). Unit: DPPH·EA, µmol Trolox equivalent (TE)/g protein; CAT, U/mg protein; GSH-PX, µmol /mg protein; T-SOD, U/mg protein. Data are presented as means ± SEM (standard error of the mean). ^a–c^ Means with no common superscripts within a group of close bars differ significantly (*p* < 0.05 or *p* < 0.01).

**Figure 2 animals-10-00311-f002:**
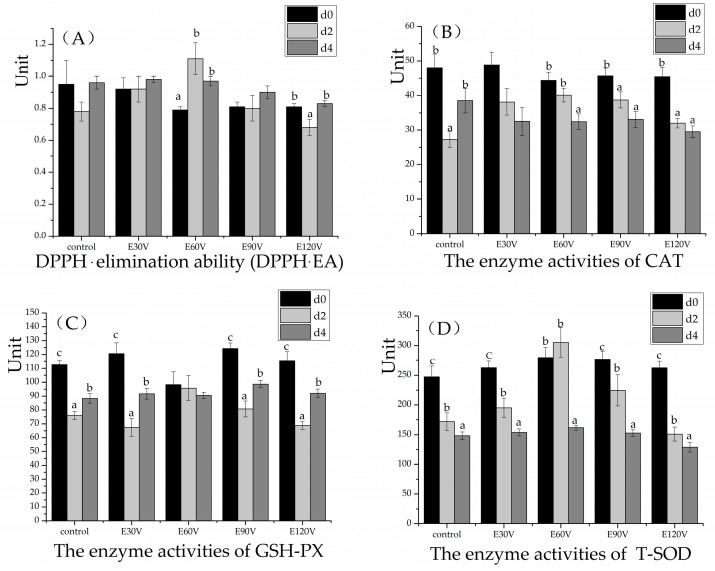
The effect of storage (4 °C) times on antioxidant capacity stability in meat goose livers with different electrical stunning (ES) methods. Figures (**A**–**D**) respectively represents the effect of ES on 1,1-diphenyl-2-picrylhydrazyl free radical (DPPH·) elimination ability (DPPH·EA), activities of catalase (CAT), glutathione peroxidase (GSH-PX), and total superoxide dismutase (T-SOD) in livers. Geese were slaughtered without stunning (the control) or after ES for 10 s with alternating current (AC) at 500 Hz in water bath which combined either with 30 V/20 mA (E30V), 60 V/40 mA (E60V), 90 V/70 mA (E90V), or with 120 V/100 mA (E120V). Livers were stored at 4 °C for 0, 2, and 4 days. Unit: DPPH· eliminating ability: µmol Trolox equivalent (TE)/g protein; CAT: U/mg protein; GSH-PX: µmol/mg protein; T-SOD: U/mg protein. Data are presented as means ± SEM (standard error of the mean). d 0: goose liver was immediately stored at −20 ± 1 °C after slaughter; d 2 and d 4: goose livers stored at 4 ± 0.5 °C for two days (48 h) and four days (96 h). ^a–c^ Means with no common superscripts within a group of close bars differ significantly (*p* < 0.05 or *p* < 0.01).

**Table 1 animals-10-00311-t001:** Composition and nutrient levels of the diet (as-fed basis).

Ingredients, %	Content
Corn	61.00
Soybean meal	25.00
Rice husk	10.40
Dicalcium phosphate	1.00
Salt	0.30
Limestone	1.20
*DL*-Methionine	0.10
Premix ^1^	1.00
Nutrient level (calculated)	
^2^ ME (MJ/kg)	10.89
^2^ CP, %	16.60
^2^ CF, %	6.63
Calcium, %	0.87
Available phosphorus, %	0.51
Lysine, %	0.81
Methionine, %	0.34

^1^ The premix was provided by the “YangDa” Feed Company (Yangzhou, China). Supplied per kilogram of the premix: retinol (1,200,000 International Unit, IU), rachitasterol (400,000 IU), D-a-tocopherol (1800 IU), coagulation vitamin (150 mg), thiamine (90 mg), riboflavin (800 mg), pyridoxine (320 mg), cobalamin (1 mg), nicotinic acid (4.5 g), pantothenic acid (1100 mg), folic acid (65 mg), biotin (5 mg), choline (45 mg), Fe (ferrous sulfate) (6 g), Cu (copper sulfate) (1 g), Mn (manganese sulfate) (9.5 g), Zn (zinc sulfate) (9 g), I (potassium iodide) (50 mg), and Se (sodium selenite) (30 mg). ^2^ ME, metabolizable energy; CP, crude protein; CF, crude fiber.

**Table 2 animals-10-00311-t002:** Effect of electrical stunning (ES) methods on malondialdehyde (MDA) in goose livers at 0~4 days postmortem (nmol/mg protein).

Time	Stunning Methods ^1^					PooledSEM	*p*-Value
	Control	E30V	E60V	E90V	E120V		
^2^ d 0	6.72 ± 0.63 ^a^ ^y^	6.29 ± 0.44 ^a^	6.87 ± 0.53 ^a^ ^xy^	8.18 ± 0.62 ^a^ ^y^	10.57 ± 0.34 ^b^ ^y^	0.41	<0.01
d 2	6.98 ± 0.62 ^y^	7.00 ± 0.86	7.60 ± 0.81 ^y^	6.00 ± 1.07 ^x^	4.34 ± 0.54 ^x^	0.38	0.06
d 4	5.09 ± 0.04 ^x^	5.41 ± 0.14	5.34 ± 0.16 ^x^	5.36 ± 0.13 ^x^	5.54 ± 0.12 ^x^	0.06	0.16
Pooled SEM	0.33	1.66	1.81	0.40	0.73		
*p*-Value	0.04	0.16	0.03	0.03	<0.01		

^1^ Geese were slaughtered without stunning (the control) or after ES for 10 s with alternating current (AC) at 500 Hz in water bath which combined either with 30 V/20 mA (E30V), 60 V/40 mA (E60V), 90 V/70 mA (E90V) or with 120 V/100 mA (E120V). Data are presented as means ± SEM (standard error of the mean). ^2^ d 0: goose liver was immediately stored at −20 ± 1 °C after slaughter; d 2 and d 4: goose livers stored at 4 ± 0.5 °C for two days (48 h) and four days (96 h). ^a-b^, ^x-y^ Means with no common superscripts within a row (a–b) or a column (x–y) differ significantly (*p* < 0.05 or *p* < 0.01)

**Table 3 animals-10-00311-t003:** Effect of electrical stunning (ES) methods on bleed-out efficiency and goose liver color on day 0 ^#^.

Variables	Stunning Methods ^2^					PooledSEM	*p*-Value
	Control	E30V	E60V	E90V	E120V		
Blood loss ^1^ (%)	5.85 ± 0.25	5.39 ± 0.09	5.41 ± 0.19	5.51 ± 0.21	5.30 ± 0.22	0.09	0.36
Lightness (L *)	27.78 ± 0.44 ^a^	29.80 ± 0.60 ^bc^	31.56 ± 0.68 ^c^	28.08 ± 0.80 ^ab^	29.95 ± 0.61 ^bc^	0.35	< 0.01
Redness (a *)	9.22 ± 0.37	10.35 ± 0.55	9.43 ± 0.28	10.40 ± 0.69	9.82 ± 0.45	0.22	0.34
Yellowness (b *)	7.37 ± 0.38	8.74 ± 0.35	9.20 ± 0.51	8.24 ± 0.47	9.05 ± 0.64	0.23	0.07

^1^Percentage (%) of body weight. ^2^ Geese were slaughtered without stunning (the control) or after ES for 10 s with alternating current (AC) at 500 Hz in water bath which combined either with 30 V/20 mA (E30V), 60 V/40 mA (E60V), 90 V/70 mA (E90V) or with 120 V/100 mA (E120V). Data are presented as means ± SEM (standard error of the mean). ^#^ Day 0 (d 0) means goose liver was immediately stored at −20 ± 1 °C after slaughter. ^a–c^ Means with no common superscripts within a row differ significantly (*p* < 0.05 or *p* < 0.01). L* means lightness, a* means redness, b* means yellowness.

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
