# Peer review of "Evaluation of the Effects of Pre-Slaughter High-Frequency Electrical Stunning Current Intensities on Lipid Oxidative Stability and Antioxidant Capacity in the Liver of Yangzhou Goose (Anser cygnoides domesticus)"

_animals, 2020, doi:10.3390/ani10020311_

Round 1

Reviewer 1 Report

Please check typing mistakes, for example, line 135 within3 months.

Author Response

Dear reviewer,

Authors are very grateful for your precious time, careful and efficient review and comment for our manuscript (manuscript ID: animals-719797)! The quality of our paper has been obviously improved due to your very valuable comments and suggestions! 

Revisions have been down according to your comments. Besides, the whole manuscript has been carefully checked again, and some extra revisions were made. Revisions that we did were highlighted in the text. If you have any questions, please don’t hesitate to contact me. Thank you very much!

Response (Res) to the reviewer

Please check typing mistakes, for example, line 135 within3 months.

Res: Thank you for this valuable feedback. We have checked the whole text and corrected these typing and grammar mistakes according to your suggestion. Revisions that we did were highlighted yellow in the text.

According to the opinions of other reviewers, other revisions have been made as follows:

Reviewer 2

Lines 185 - 186. Rephrase this sentence “The MDA level in all of these groups decreased at d 4 as compared with d 0 except for the E30V and E60V group (P< 0.05) in the present study.”

E.g., The MDA level in all of these groups decreased at d 4 as compared with d 0 (P < 0.05), except for the E30V and E60V group (P > 0.05) in the present study.

Res: We have rephrased this sentence in lines 187 - 188 according to your suggestion.

Lines 208 - 210: Rephrase this sentence “The enzyme activities of CAT in control and E120V groups, the enzyme activities of GSH-PX in control, E30V and E120V groups was lower than that in E60V group (P< 0.05, Figure 1B).”

E.g., The enzyme activities values of CAT in control and E120V groups were lower (P < 0.05) than in E60V group, in addition, CAT in control was lower than that in E30 and E90 (< 0.05). The enzyme activities values of GSH-PX in control, E30V and E120V groups were lower compared to that from E60V (Figure 1B).

Res: We have rephrased this sentence in lines 210 - 213 according to your suggestion.

Lines 201 - 211: Rephrase this sentence “The enzyme activities of T-AOC in E60V was the highest, and had significant difference with that in other groups (P< 0.05, Figure 1B).”

E.g. The enzyme activities of T-AOC in E60V was higher compared to those from other groups (P < 0.05, Figure 1B).

Res: We have rephrased this sentence in lines 213 - 215 according to your suggestion.

Reviewer 3

The Authors revised their paper according to the comments and suggestions of reviewers; so, in my opinion, the revised paper merits the final acceptance.

Res: Thank you very much for your review and comments! The whole manuscript has been carefully checked again, and minor revisions were made and highlighted in the revised manuscript.

Extra revisions done by authors

Most of the Arabic numbers that are less than ten were revised as words (e.g., 4 to four) throughout the text. The sentences in the notes of each figure (Fig. 1, Fig. 2) were combined into one paragraph, respectively. We have also made extra revisions to make the sentences clearer without changing the original meaning of the sentences. Revisions that we did were highlighted yellow in the text, e.g., lines 114-116, 237 - 238 and 242 - 244.

Once again, authors appreciate for your comments and suggestions!

Best regards!

Sincerely yours,

Xin Zhang, 

College of Animal Science and Technology, Yangzhou University

Yangzhou City, Jiangsu Province, P. R. China, 225009

Reviewer 2 Report

Lines 185-186. Rephrase this sentence “The MDA level in all of these groups decreased at d 4 as compared with d 0 except for the E30V 185 and E60V group (P < 0.05) in the present study.”

E.g: “The MDA level in all of these groups decreased at d 4 as compared with d 0 (P < 0.05), except for the E30V 185 and E60V group (P > 0.05) in the present study.

Lines 208-210: Rephrase this sentence “The enzyme activities of CAT in control and E120V groups, the enzyme activities of GSH-PX in control, E30V and E120V groups was lower than that in E60V group (P < 0.05, Figure 1B).”

E.g.: The enzyme activities values of CAT in control and E120V groups were lower (P < 0.05) than in E60V group, in addition, CAT in control was lower than that in E30 and E90 (P < 0.05). The enzyme activities values of GSH-PX in control, E30V and E120V groups were lower compared to that from E60V (Figure 1B).”

Lines 201-211: Rephrase this sentence “The enzyme activities of T-AOC in E60V was the highest, and had significant difference with that in other groups (P < 0.05, Figure 1B).”

E.g. The enzyme activities of T-AOC in E60V was higher compared to those from other groups (P < 0.05, Figure 1B).

Author Response

Dear reviewer,

Authors are very grateful for your precious time, careful and efficient review and comment for our manuscript (manuscript ID: animals-719797)! The quality of our paper has been obviously improved due to your very valuable comments and suggestions! 

Revisions have been down according to your comments. Besides, the whole manuscript has been carefully checked again, and some extra revisions were made. Revisions that we did were highlighted in the text. If you have any questions, please don’t hesitate to contact me. Thank you very much!

Response (Res) to the reviewer

Lines 185 - 186. Rephrase this sentence “The MDA level in all of these groups decreased at d 4 as compared with d 0 except for the E30V and E60V group (P< 0.05) in the present study.”

E.g., The MDA level in all of these groups decreased at d 4 as compared with d 0 (P < 0.05), except for the E30V and E60V group (P > 0.05) in the present study.

Res: We have rephrased this sentence in lines 187 - 188 according to your suggestion.

Lines 208 - 210: Rephrase this sentence “The enzyme activities of CAT in control and E120V groups, the enzyme activities of GSH-PX in control, E30V and E120V groups was lower than that in E60V group (P< 0.05, Figure 1B).”

E.g., The enzyme activities values of CAT in control and E120V groups were lower (P < 0.05) than in E60V group, in addition, CAT in control was lower than that in E30 and E90 (< 0.05). The enzyme activities values of GSH-PX in control, E30V and E120V groups were lower compared to that from E60V (Figure 1B).

Res: We have rephrased this sentence in lines 210 - 213 according to your suggestion.

Lines 201 - 211: Rephrase this sentence “The enzyme activities of T-AOC in E60V was the highest, and had significant difference with that in other groups (P< 0.05, Figure 1B).”

E.g. The enzyme activities of T-AOC in E60V was higher compared to those from other groups (P < 0.05, Figure 1B).

Res: We have rephrased this sentence in lines 213 - 215 according to your suggestion.

According to the opinions of other reviewers, other revisions have been made as follows:

Reviewer 2

Please check typing mistakes, for example, line 135 within3 months.

Res: Thank you for this valuable feedback. We have checked the whole text and corrected these typing and grammar mistakes according to your suggestion. Revisions that we did were highlighted yellow in the text.

Reviewer 3

The Authors revised their paper according to the comments and suggestions of reviewers; so, in my opinion, the revised paper merits the final acceptance.

Res: Thank you very much for your review and comments! The whole manuscript has been carefully checked again, and minor revisions were made and highlighted in the revised manuscript.

Extra revisions done by authors

Most of the Arabic numbers that are less than ten were revised as words (e.g., 4 to four) throughout the text. The sentences in the notes of each figure (Fig. 1, Fig. 2) were combined into one paragraph, respectively. We have also made extra revisions to make the sentences clearer without changing the original meaning of the sentences. Revisions that we did were highlighted yellow in the text, e.g., lines 114-116, 237 - 238 and 242 - 244.

Once again, authors appreciate for your comments and suggestions!

Best regards!

Sincerely yours,

Xin Zhang, 

College of Animal Science and Technology, Yangzhou University

Yangzhou City, Jiangsu Province, P. R. China, 225009

Reviewer 3 Report

The Authors revised their paper according the comments and suggestions of reviewers; so, in my opinion the revised paper merits the final acceptance.

Author Response

Dear reviewer,

Authors are very grateful for your precious time, careful and efficient review and comment for our manuscript (manuscript ID: animals-719797)! The quality of our paper has been obviously improved due to your very valuable comments and suggestions! 

The whole manuscript has been carefully checked again, and some extra revisions were made. Revisions that we did were highlighted in the text. If you have any questions, please don’t hesitate to contact me. Thank you very much!

Response (Res) to the reviewer

The Authors revised their paper according to the comments and suggestions of reviewers; so, in my opinion, the revised paper merits the final acceptance.

Res: Thank you very much for your review and comments! The whole manuscript has been carefully checked again, and minor revisions were made and highlighted in the revised manuscript.

According to the opinions of other reviewers, other revisions have been made as follows:

Reviewer 2

Please check typing mistakes, for example, line 135 within3 months.

Res: Thank you for this valuable feedback. We have checked the whole text and corrected these typing and grammar mistakes according to your suggestion. Revisions that we did were highlighted yellow in the text.

Reviewer 3

Lines 185 - 186. Rephrase this sentence “The MDA level in all of these groups decreased at d 4 as compared with d 0 except for the E30V and E60V group (P< 0.05) in the present study.”

E.g., The MDA level in all of these groups decreased at d 4 as compared with d 0 (P < 0.05), except for the E30V and E60V group (P > 0.05) in the present study.

Res: We have rephrased this sentence in lines 187 – 188 according to your suggestion.

Lines 208 - 210: Rephrase this sentence “The enzyme activities of CAT in control and E120V groups, the enzyme activities of GSH-PX in control, E30V and E120V groups was lower than that in E60V group (P< 0.05, Figure 1B).”

E.g., The enzyme activities values of CAT in control and E120V groups were lower (P < 0.05) than in E60V group, in addition, CAT in control was lower than that in E30 and E90 (< 0.05). The enzyme activities values of GSH-PX in control, E30V and E120V groups were lower compared to that from E60V (Figure 1B).

Res: We have rephrased this sentence in lines 210 - 213 according to your suggestion.

Lines 201 - 211: Rephrase this sentence “The enzyme activities of T-AOC in E60V was the highest, and had significant difference with that in other groups (P< 0.05, Figure 1B).”

E.g. The enzyme activities of T-AOC in E60V was higher compared to those from other groups (P < 0.05, Figure 1B).

Res: We have rephrased this sentence in lines 213 - 215 according to your suggestion.

Extra revisions done by authors

Most of the Arabic numbers that are less than ten were revised as words (e.g., 4 to four) throughout the text. The sentences in the notes of each figure (Fig. 1, Fig. 2) were combined into one paragraph, respectively. We have also made extra revisions to make the sentences clearer without changing the original meaning of the sentences. Revisions that we did were highlighted yellow in the text, e.g., lines 114-116, 237 - 238 and 242 – 244.

Once again, authors appreciate for your comments and suggestions!

Best regards!

Sincerely yours,

Xin Zhang, 

College of Animal Science and Technology, Yangzhou University

Yangzhou City, Jiangsu Province, P. R. China, 225009

This manuscript is a resubmission of an earlier submission. The following is a list of the peer review reports and author responses from that submission.

Round 1

Reviewer 1 Report

Lines 9-16: There are some mistakes according to Instructions for the authors (Correspondence? don’t need telephone numbers?)

Line 40: Use the same aberration in the whole text (DPPH•).

Line 44: Add liver or meat goose liver as a keyword (liver, colour…)

Line 47, Introduction: There is nothing about storage time of goose liver in connection with oxidative stress in the introduction, add some researches about this topic. There are no published data about liver colour. Way is this important, for customers?

Lines 81-82: What about storage time and its effects on…. please add. Can you write the aim in the same order than later in materials and methods, results… (colour of the liver is at the end)?

Lines 85-87: This is sentence is a part of for Introduction, because describe Yangzhou geese and it isn’t describing the Materials and methods of the present research.

Line 94: Table 1., You don’t need Total (100.00). Can you explain what ME, CP and CF mean?

Line 129-133: What about weights of liver (W1) – please add in the results. I can’t find this numbers (Blood loss) in the results.

Line 134, chapter 2.4. Please add how long till analysis of lipid oxidation and antioxidant capacity samples were stored.

Lines 143; 145: GSH-Px or GSH-PX; use the same abbreviation in the text

Line 148: The 1, 1-Diphenyl-2-picrylhydrazyl (DPPH ), is D small letter

Line 167 – 173: I don’t understand? Materials and methods should…. This is text from the Article Template!

Line 199: . is missing at the end of the table title. What does prot mean = protein? Add in the title of the table malondialdehyde (MDA) concentration. Can you explain (in materials and methods) how you calculate MDA concentrations/mg proteins.

Line 200: Table 2. Can you explain below the table what d 0, d 2 and d 4 mean?

Line 208: DPPH• elimination ability, above you used abbreviation (DPPH) ?

Line 223, Figure 1: A, B, C up and down in the figure? There isn’t the same size and shape of all three figures and also letters. Do you really need on all figures name Antioxidative capacity?

Line 251, Figure 2: A, B, C up and down in the figure? Do you really need on all figures, name Electrical stunning methods? What mean prot (protein or aberration?)

Line 257: There is a lot of space … Livers…and ….C

Lines 283 -286: I don’t understand the text Authors should discuss the results and how they

can be interpreted in perspective of previous studies and of the working hypotheses. The findings and their implications should be discussed in the broadest context possible. Future research

directions may also be highlighted. This is text from the Article Template!

Line 287. Dot is missing at the end of the table title.

Why extra P < 0.01 in the Lightness (second row), P-value is in the firs row in the table?

Line 288. Dot is missing at the end of the sentence.

Line 293 Conclusions; Rewrite the conclusions in order of the materials and methods and results, lipid oxidation….colour. Add what these results suggest; which combination could be applied in the ES to improve the lipid oxidative stability…

Lines 300-301. There are not space between the letter of the names. Please check the Article template and Instructions for the Authors in text.

Author Response

Dear editors and reviewers,

Authors are very grateful for your precious time, careful and efficient review and comments for our manuscript (manuscript ID: animals-679198)! Your comments are very valuable for improving the quality of our paper! Revisions have been down according to your suggestions. Besides, the whole manuscript has been carefully checked again, and some extra revisions were made. These main changes are listed at the end of this response letter. Revisions that we did are highlighted in the text. If you have any questions, please don’t hesitate to contact me. Thank you very much!

Response (Res) to reviewer

Lines 9-16: There are some mistakes according to Instructions for the authors (Correspondence? don’t need telephone numbers?) 

Res: â‘ We have changed the “Corresponding author:” with “Correspondence” in line 15.

â‘¡We have provided the telephone number in line 15, but the format incorrect. We have correct it already in line 15.

Line 40: Use the same aberration in the whole text (DPPH•). 

Res: We have changed the “DPPH•” with “DPPH•” in the whole text, and highlighted them in yellow.

Line 44: Add liver or meat goose liver as a keyword (liver, colour…)

Res: We have added “liver” in the keywords and replaced “liver colour” with “colour” in line 40. 

Line 47: Introduction: There is nothing about storage time of goose liver in connection with oxidative stress in the Introduction, add some researches about this topic. There are no published data about liver colour. The way is this important, for customers?

Res: â‘ We think that there is no connection between storage time and oxidative stress (it refers to a state of live animals). Do you mean the connection between storage time and lipid oxidation? We have added one reference about the relationship between storage time and oxidative stress in lines 59 - 61. We have add “live” before “birds” in line 58.

â‘¡Goose liver has a large number of consumers in the market of China. Lipid oxidation products may damage the health of consumers. Liver colour is also an important factor in judging lipid oxidation and freshness of the liver.

Lines 81-82: What about storage time and its effects on…. please add. Can you write the aim in the same order than later in materials and methods, results… (colour of the liver is at the end)?

Res: â‘ The “storage time and its effects on…” has been added into the aim in lines 80 - 81.

â‘¡We have put the “colour” after antioxidant capacity in line 81.

Lines 85-87: This is sentence is a part of for Introduction, because describe Yangzhou geese and it isn’t describing the Materials and methods of the present research.

Res: This sentence was a duplicate of the first sentence in Introduction. We have deleted this sentence and its citation from the Material & Method. In addition, the reference was also deleted from the Reference section. 

Line 94: Table 1., You don’t need Total (100.00). Can you explain what ME, CP and CF mean?

Res: We have deleted the Total (100.00), and added the explanation of ME, CP and CF in line 100.

Line 129-133: What about weights of the liver (W1) – please add in the results. I can’t find these numbers (Blood loss) in the results.

Res: â‘ The W1 referred to geese weight rather than liver weight. We didn’t weigh the liver.

â‘¡The numbers of blood loss were in table 3.

Line 134, chapter 2.4. Please add how long till analysis of lipid oxidation and antioxidant capacity samples were stored. 

Res: We measured those variables within three months after sampling. We have added the information in lines 135 - 137.

Lines 143; 145: GSH-Px or GSH-PX; use the same abbreviation in the text.

Res: We have changed the “GSH-Px” with “GSH-PX” in the whole text, and highlighted them in yellow.

Line 148: The 1, 1-Diphenyl-2-picrylhydrazyl (DPPH •), is D small letter.

Res: Yes, it should be a small letter. We have revised “1, 1-Diphenyl-2-picrylhydrazyl” to “1, 1-diphenyl-2-picrylhydrazyl” in line 149.

Line 167 – 173: I don’t understand? Materials and methods should…. This is the text from the Article Template!

Res: Yes, these sentences came from the Article Template. Sorry, it was our mistake. We have deleted these words of the template after line 169.

Line 199: .is missing at the end of the table title. What does prot mean = protein? Add in the title of the table malondialdehyde (MDA) concentration. Can you explain (in materials and methods) how you calculate MDA concentrations/mg proteins.

Res: â‘ We have added the dot at the end of the table title in line 93, 196 and 290. 

â‘¡The “prot” means protein. We have changed the “prot” with “protein” in the whole text. 

â‘¢The results of MDA concentrations were calculated on the basis of the protein content in liver homogenates. We have added the related information in lines 147 - 148.

Line 200: Table 2. Can you explain below the table what d 0, d 2 and d 4 mean?

Res: The d 0, d 2 and d 4, respectively means day 0, 2, and 4. We have added the information in lines 200 - 201.

Line 208: DPPH• elimination ability, above you used abbreviation (DPPH)?

Res: DPPH is short for 1, 1-diphenyl-2-picrylhydrazyl, whereas, DPPH• refers to DPPH free radical. The “DPPH• elimination ability” is abbreviated as DPPH•EA throughout the text of our revised manuscript.

Line 223, Figure 1: A, B, C up and down in the figure? There isn’t the same size and shape of all three figures and also letters. Do you really need on all figures name Antioxidative capacity?

Res: We have corrected and merged the photos A, B, C into one picture (Figure 1).

Line 251, Figure 2: A, B, C up and down in the figure? Do you really need on all figures, name Electrical stunning methods? What mean prot (protein or aberration?)

Res: â‘ We have corrected and merged these pictures into Figure 2. 

â‘¡We have renamed all the figures in Figure 2. 

     â‘¢The “prot” means protein. We have explained it throughout the text of the manuscript.

Line 257: There is a lot of space … Livers…and ….C

Res: We have deleted the spare space in line 255.

Lines 283 -286: I don’t understand the text Authors should discuss the results and how they can be interpreted in perspective of previous studies and of the working hypotheses. The findings and their implications should be discussed in the broadest context possible. Future research directions may also be highlighted. This is the text from the Article Template!

Res: Yes, these sentences came from the Article Template. Sorry, it was our mistake. We have deleted these words after line 286.

Line 287. Dot is missing at the end of the table title. Why extra P< 0.01 in the Lightness (second row), P-value is in the first row in the table? 

Res: â‘ We have added the dot at the end of the table title in line 290. 

â‘¡The p-value is in the last column. The P-value of brightness is less than 0.01, which cannot be filled in the form accurately, so we use “p < 0.01” instead before. We have deleted the “p” of the “p < 0.01” in the second row. A similar mistake was corrected in Table 3.c.

Line 288. Dot is missing at the end of the sentence.

Res: We have added the dot at the end of the sentence in line 291.

Line 293 Conclusions; Rewrite the conclusions in order of the materials and methods and results, lipid oxidation….colour. Add what these results suggest; which combination could be applied in the ES to improve the lipid oxidative stability.

Res: â‘ We have revised “the lightness, antioxidant capacity and lipid oxidative stability” to “the antioxidant capacity, lipid oxidative stability and the lightness” in lines 298 - 299.

â‘¡We have added the suggestion of results in lines 300-301.

Lines 300-301. There are not space between the letter of the names. Please check the Article template and Instructions for the Authors in text. 

Res: Yes, we have deleted the spare spaces in lines 304 - 305.

According to the opinions of other reviewers, there have been revised as follows.

Reviewer 1

The Authors have investigated an interesting topic and the theme has been properly described. I would like to congratulate authors for the good-quality of the article, the literature reported used to write the paper, and for the clear and appropriate structure. The manuscript is well written, presented and discussed, and understandable to a specialist readership. In general, the organization and the structure of the article are satisfactory and in agreement with the journal instructions for authors. The subject is adequate with the overall journal scope. The work shows a conscientious study in which a very exhaustive discussion of the literature available has been carried out. The introduction provides sufficient background, and the other sections include results clearly presented and analyzed exhaustively. As specific comment, the Simple Summary could be shortened. 

Res: Thank you for your suggestion. We have shortened the Simple Summary. Details are listed as followings:

â‘ We have replaced the “However, limited research has been performed to evaluate the ES effects on the liver lipid oxidative stability of the meat geese.” with “Limited research has been performed on the effect of ES on liver quality of the meat geese” in lines 19 - 20.

â‘¡We have deleted these words: “Geese were exposed to ES for 10 s with alternating current (AC) at 500 Hz in a water bath, which was combined with one of four different current intensities”. 

â‘¢ We have replaced “Stunning each goose with the current intensity at 60 V/40 mA improved the lightness, antioxidant capacity, and lipid oxidative stability in livers of meat geese during cold storage from day 0 to day four as compared with high-current ES (120 V/100 mA) based on the same high electrical frequency (500 Hz, AC).” with “Stunning each goose with the current intensity at 60 V/40 mA improved antioxidant capacity and lipid oxidative stability from day 0 to day four and the lightness at day 0 in livers of meat geese during cold storage.” in lines 22 - 24.

â‘£We have replaced “the lipid oxidative stability and antioxidant capacity” with “the antioxidant capacity and lipid oxidative stability.” in lines 25 - 26 through out the text according to the suggestion of other reviewers.

Reviewer 2

Title

change “livers” with “liver”.

Res: We have changed “the livers of Yangzhou geese” with “the liver of Yangzhou goose” in the whole text, and highlighted them in yellow. 

Simple Summary:

Line 23: “meet”?

Res: We have revised “meet” to “meat” in line 21.

Introduction

Line 67. “animal welfare” Generally animal welfare is used for living animals. Please change with other words.

Res: We have replaced “animal welfare” with “reduce the pain of poultry” in lines 64 - 65.

M&M

Lines 85-87: Move this sentences in introduction section.

Res: We have deleted the introduction of Yangzhou geese in the material method, and make the corresponding modification in the reference of the article. 

Lines 87-91: I suggest to include some information regarding the rearing condition. In addition: the animals had free access to the water?

Res: We have added the information of rearing condition in lines 84 - 88.

Line 129: were the geese subjected to the fasting period?

Res: All the geese were weighed individually after fasting for 6 h. We have added the information in line 128.

Lines 168- 173: What does this part mean? What do the authors want to describe? It is not clear.

Res: Sorry, it is Article Template. We have deleted these words of the template after line 169.

R&D

Line 180: change “P< 0.01” with “P < 0.05” (the differences reported among means values, shown in the table 2, are for P < 0.05_ a-b).

Res: The “P < 0.05” is the cut off level for significant difference. The “< 0.01” indicates highly significant. We didn't write it clearly before. We have added this information throughout the text (Statistic analysis, Tables, and Figures). We have not changed "P < 0.01" with "P < 0.05" in line 176.

Lines 185-186: replace “&” with “and”.

Res: We have replaced “&” with “and” in line 182.

Lines 189-190: This sentence does not describe correctly the results regarding the effect of the storage time on MDA content. E60V group doesn’t show statistical difference according to with the letters x and y. Rewrite the sentence.

Res: We have revised the sentence in line187.

Line 194: “The results suggested that …” replace with “The results of the present study suggested that …”

Res: We have replaced: “The results suggested that …” with “The results of the present study suggested that …” in lines 191 - 192. 

Lines 207-209: the sentence “However, at 2 d postmortem, the DPPH elimination ability, the enzyme activities of CAT, GSH-PX and T-SOD in geese livers was higher in the E60V group than in E120V groups (P< 0.05, Fig. 1B).” does not describe the results correctly from Fig. 1B (e.g. DPPH: E 30V versus E120V. CAT: Control versus E60V. T-SOD: E90V versus E120V; E60V versus E90V).

Res: We have rewritten the sentence in lines 208 - 212.

Lines 214-216: revise and complete this sentence “At 4 d postmortem, the DPPH elimination ability, the activity of T-SOD in the livers of geese was higher in the E60V groups than in the E120V group (P< 0.05, Fig. 1C), according to with the results reported in Fig 1C.

Res: We have rewritten the sentence in lines 217 - 220.

Lines 262-264: All the differences showed in table 3 are for P< 0.05 (a-c). Why the authors reported the Pvalue effect of ANOVA < 0.01 and P = 0.07? In addition, others significant differences (P < 0.05) among groups studied were not reported. Check carefully.

Res: â‘ The “P < 0.05” is the cut off level for “significant difference. The “< 0.01” indicates highly significant. We have added this information throughout the text (Statistic analysis, Tables, and Figures).

â‘¡ We have added the other significant results which have significant differences in lines 265 - 266. 

Lines 280-281: the following sentence is not correct: “In the present study, the decrease of lightness and yellowness in E120V was consistent with a higher MDA level at d 0.”.

Res: We have changed “decrease” to “increase” in line 283.

Lines 283-286: these sentences are not clear for me:. “Authors should discuss the results and how they can be interpreted in perspective of previous studies and of the working hypotheses. The findings and their implications should be discussed in the broadest context possible. Future research directions may also be highlighted.” What the authors want to say? Is it a mistake?

Res: Sorry, it is the text from the Article Template. We have deleted these words of the template after line 286.

Table 3

Title: report in the title that the data regard time 0.

Res: We have revised the table title in lines 289-290.

Figure 1 and 2: include “means ± <SD”

Res: We have added the information in line 232 and 259.

Extra revisions done by authors.

We have revised the grammar of the whole text, and highlighted them in yellow. We have add “improve”before “product” in line 65. We have add “the”before “the” in line 65. We have abbreviated “DPPHelimination ability” to “DPPH•EA”, and revised in the whole text. We have revisedthe thickness of horizontal line in tables. We have add “that in”after “than” in lines 176, 267 and 269. We have revised“get touch with” to “touch” in line 113. We have add “ (TE)”after “Trolox equivalent” in lines 157. We have add “be”after “may” in lines 184.

Once again, thank you very much for your comments and suggestions.

Best regards,

Sincerely yours,

Xin Zhang, 

College of Animal Science and Technology, Yangzhou University

Yangzhou City, Jiangsu Province, P. R. China, 225009

Reviewer 2 Report

The Authors have investigated an interesting topic and the theme has been properly described. I would like to congratulate authors for the good-quality of the article, the literature reported used to write the paper, and for the clear and appropriate structure. The manuscript is well written, presented and discussed, and understandable to a specialist readership. In general, the organization and the structure of the article are satisfactory and in agreement with the journal instructions for authors. The subject is adequate with the overall journal scope. The work shows a conscientious study in which a very exhaustive discussion of the literature available has been carried out. The introduction provides sufficient background, and the other sections include results clearly presented and analyzed exhaustively. As specific comment, the Simple Summary could be shortened. 

So, I recommend the acceptance of the paper.

Author Response

Dear editors and reviewers,

   Authors are very grateful for your precious time, careful and efficient review and comments for our manuscript (manuscript ID: animals-679198)! Your comments are very valuable for improving the quality of our paper! Revisions have been down according to your suggestions. Besides, the whole manuscript has been carefully checked again, and some extra revisions were made. These main changes are listed at the end of this response letter. Revisions that we did are highlighted in the text. If you have any questions, please don’t hesitate to contact me. Thank you very much!

Response (Res) to reviewer

The Authors have investigated an interesting topic and the theme has been properly described. I would like to congratulate authors for the good-quality of the article, the literature reported used to write the paper, and for the clear and appropriate structure. The manuscript is well written, presented and discussed, and understandable to a specialist readership. In general, the organization and the structure of the article are satisfactory and in agreement with the journal instructions for authors. The subject is adequate with the overall journal scope. The work shows a conscientious study in which a very exhaustive discussion of the literature available has been carried out. The introduction provides sufficient background, and the other sections include results clearly presented and analyzed exhaustively. As specific comment, the Simple Summary could be shortened. 

Res: Thank you for your suggestion. We have shortened the Simple Summary. Details are listed as followings:

â‘ We have replaced the “However, limited research has been performed to evaluate the ES effects on the liver lipid oxidative stability of the meat geese.” with “Limited research has been performed on the effect of ES on liver quality of the meat geese” in lines 19 - 20.

â‘¡We have deleted these words: “Geese were exposed to ES for 10 s with alternating current (AC) at 500 Hz in a water bath, which was combined with one of four different current intensities”. 

â‘¢ We have replaced “Stunning each goose with the current intensity at 60 V/40 mA improved the lightness, antioxidant capacity, and lipid oxidative stability in livers of meat geese during cold storage from day 0 to day four as compared with high-current ES (120 V/100 mA) based on the same high electrical frequency (500 Hz, AC).” with “Stunning each goose with the current intensity at 60 V/40 mA improved antioxidant capacity and lipid oxidative stability from day 0 to day four and the lightness at day 0 in livers of meat geese during cold storage.” in lines 22 - 24.

â‘£We have replaced “the lipid oxidative stability and antioxidant capacity” with “the antioxidant capacity and lipid oxidative stability.” in lines 25 - 26 through out the text according to the suggestion of other reviewers.

According to the opinions of other reviewers, there have been revised as follows.

Reviewer 1

Lines 9-16: There are some mistakes according to Instructions for the authors (Correspondence? don’t need telephone numbers?) 

Res: â‘ We have changed the “Corresponding author:” with “Correspondence” in line 15.

â‘¡We have provided the telephone number in line 15, but the format incorrect. We have correct it already in line 15.

Line 40: Use the same aberration in the whole text (DPPH•). 

Res: We have changed the “DPPH•” with “DPPH•” in the whole text, and highlighted them in yellow.

Line 44: Add liver or meat goose liver as a keyword (liver, colour…)

Res: We have added “liver” in the keywords and replaced “liver colour” with “colour” in line 40. 

Line 47: Introduction: There is nothing about storage time of goose liver in connection with oxidative stress in the Introduction, add some researches about this topic. There are no published data about liver colour. The way is this important, for customers?

Res: â‘ We think that there is no connection between storage time and oxidative stress (it refers to a state of live animals). Do you mean the connection between storage time and lipid oxidation? We have added one reference about the relationship between storage time and oxidative stress in lines 59 - 61. We have add “live” before “birds” in line 58.

â‘¡Goose liver has a large number of consumers in the market of China. Lipid oxidation products may damage the health of consumers. Liver colour is also an important factor in judging lipid oxidation and freshness of the liver.

Lines 81-82: What about storage time and its effects on…. please add. Can you write the aim in the same order than later in materials and methods, results… (colour of the liver is at the end)?

Res: â‘ The “storage time and its effects on…” has been added into the aim in lines 80 - 81.

â‘¡We have put the “colour” after antioxidant capacity in line 81.

Lines 85-87: This is sentence is a part of for Introduction, because describe Yangzhou geese and it isn’t describing the Materials and methods of the present research.

Res: This sentence was a duplicate of the first sentence in Introduction. We have deleted this sentence and its citation from the Material & Method. In addition, the reference was also deleted from the Reference section. 

Line 94: Table 1., You don’t need Total (100.00). Can you explain what ME, CP and CF mean?

Res: We have deleted the Total (100.00), and added the explanation of ME, CP and CF in line 100.

Line 129-133: What about weights of the liver (W1) – please add in the results. I can’t find these numbers (Blood loss) in the results.

Res: â‘ The W1 referred to geese weight rather than liver weight. We didn’t weigh the liver.

â‘¡The numbers of blood loss were in table 3.

Line 134, chapter 2.4. Please add how long till analysis of lipid oxidation and antioxidant capacity samples were stored. 

Res: We measured those variables within three months after sampling. We have added the information in lines 135 - 137.

Lines 143; 145: GSH-Px or GSH-PX; use the same abbreviation in the text.

Res: We have changed the “GSH-Px” with “GSH-PX” in the whole text, and highlighted them in yellow.

Line 148: The 1, 1-Diphenyl-2-picrylhydrazyl (DPPH •), is D small letter.

Res: Yes, it should be a small letter. We have revised “1, 1-Diphenyl-2-picrylhydrazyl” to “1, 1-diphenyl-2-picrylhydrazyl” in line 149.

Line 167 – 173: I don’t understand? Materials and methods should…. This is the text from the Article Template!

Res: Yes, these sentences came from the Article Template. Sorry, it was our mistake. We have deleted these words of the template after line 169.

Line 199: .is missing at the end of the table title. What does prot mean = protein? Add in the title of the table malondialdehyde (MDA) concentration. Can you explain (in materials and methods) how you calculate MDA concentrations/mg proteins.

Res: â‘ We have added the dot at the end of the table title in line 93, 196 and 290. 

â‘¡The “prot” means protein. We have changed the “prot” with “protein” in the whole text. 

â‘¢The results of MDA concentrations were calculated on the basis of the protein content in liver homogenates. We have added the related information in lines 147 - 148.

Line 200: Table 2. Can you explain below the table what d 0, d 2 and d 4 mean?

Res: The d 0, d 2 and d 4, respectively means day 0, 2, and 4. We have added the information in lines 200 - 201.

Line 208: DPPH• elimination ability, above you used abbreviation (DPPH)?

Res: DPPH is short for 1, 1-diphenyl-2-picrylhydrazyl, whereas, DPPH• refers to DPPH free radical. The “DPPH• elimination ability” is abbreviated as DPPH•EA throughout the text of our revised manuscript.

Line 223, Figure 1: A, B, C up and down in the figure? There isn’t the same size and shape of all three figures and also letters. Do you really need on all figures name Antioxidative capacity?

Res: We have corrected and merged the photos A, B, C into one picture (Figure 1).

Line 251, Figure 2: A, B, C up and down in the figure? Do you really need on all figures, name Electrical stunning methods? What mean prot (protein or aberration?)

Res: â‘ We have corrected and merged these pictures into Figure 2. 

        â‘¡We have renamed all the figures in Figure 2. 

        â‘¢The “prot” means protein. We have explained it throughout the text of the manuscript.

Line 257: There is a lot of space … Livers…and ….C

Res: We have deleted the spare space in line 255.

Lines 283 -286: I don’t understand the text Authors should discuss the results and how they can be interpreted in perspective of previous studies and of the working hypotheses. The findings and their implications should be discussed in the broadest context possible. Future research directions may also be highlighted. This is the text from the Article Template!

Res: Yes, these sentences came from the Article Template. Sorry, it was our mistake. We have deleted these words after line 286.

Line 287. Dot is missing at the end of the table title. Why extra P< 0.01 in the Lightness (second row), P-value is in the first row in the table? 

Res: â‘ We have added the dot at the end of the table title in line 290. 

â‘¡The p-value is in the last column. The P-value of brightness is less than 0.01, which cannot be filled in the form accurately, so we use “p < 0.01” instead before. We have deleted the “p” of the “p < 0.01” in the second row. A similar mistake was corrected in Table 3.c.

Line 288. Dot is missing at the end of the sentence.

Res: We have added the dot at the end of the sentence in line 291.

Line 293 Conclusions; Rewrite the conclusions in order of the materials and methods and results, lipid oxidation….colour. Add what these results suggest; which combination could be applied in the ES to improve the lipid oxidative stability.

Res: â‘ We have revised “the lightness, antioxidant capacity and lipid oxidative stability” to “the antioxidant capacity, lipid oxidative stability and the lightness” in lines 298 - 299.

â‘¡We have added the suggestion of results in lines 300-301.

Lines 300-301. There are not space between the letter of the names. Please check the Article template and Instructions for the Authors in text. 

Res: Yes, we have deleted the spare spaces in lines 304 - 305.

Reviewer 2

Title: 

change “livers” with “liver”.

Res: We have changed “the livers of Yangzhou geese” with “the liver of Yangzhou goose” in the whole text, and highlighted them in yellow. 

Simple Summary:

Line 23: “meet”?

Res: We have revised “meet” to “meat” in line 21.

Introduction: 

Line 67. “animal welfare” Generally animal welfare is used for living animals. Please change with other words.

Res: We have replaced “animal welfare” with “reduce the pain of poultry” in lines 64 - 65.

M&M

Lines 85-87: Move this sentences in introduction section.

Res: We have deleted the introduction of Yangzhou geese in the material method, and make the corresponding modification in the reference of the article. 

Lines 87-91: I suggest to include some information regarding the rearing condition. In addition: the animals had free access to the water?

Res: We have added the information of rearing condition in lines 84 - 88.

Line 129: were the geese subjected to the fasting period?

Res: All the geese were weighed individually after fasting for 6 h. We have added the information in line 128.

Lines 168- 173: What does this part mean? What do the authors want to describe? It is not clear.

Res: Sorry, it is Article Template. We have deleted these words of the template after line 169.

R&D

Line 180: change “P< 0.01” with “P < 0.05” (the differences reported among means values, shown in the table 2, are for P < 0.05_ a-b).

Res: The “P < 0.05” is the cut off level for significant difference. The “< 0.01” indicates highly significant. We didn't write it clearly before. We have added this information throughout the text (Statistic analysis, Tables, and Figures). We have not changed "P < 0.01" with "P < 0.05" in line 176.

Lines 185-186: replace “&” with “and”.

Res: We have replaced “&” with “and” in line 182.

Lines 189-190: This sentence does not describe correctly the results regarding the effect of the storage time on MDA content. E60V group doesn’t show statistical difference according to with the letters x and y. Rewrite the sentence.

Res: We have revised the sentence in line187.

Line 194: “The results suggested that …” replace with “The results of the present study suggested that …”

Res: We have replaced: “The results suggested that …” with “The results of the present study suggested that …” in lines 191 - 192. 

Lines 207-209: the sentence “However, at 2 d postmortem, the DPPH elimination ability, the enzyme activities of CAT, GSH-PX and T-SOD in geese livers was higher in the E60V group than in E120V groups (P< 0.05, Fig. 1B).” does not describe the results correctly from Fig. 1B (e.g. DPPH: E 30V versus E120V. CAT: Control versus E60V. T-SOD: E90V versus E120V; E60V versus E90V).

Res: We have rewritten the sentence in lines 208 - 212.

Lines 214-216: revise and complete this sentence “At 4 d postmortem, the DPPH elimination ability, the activity of T-SOD in the livers of geese was higher in the E60V groups than in the E120V group (P< 0.05, Fig. 1C), according to with the results reported in Fig 1C.

Res: We have rewritten the sentence in lines 217 - 220.

Lines 262-264: All the differences showed in table 3 are for P< 0.05 (a-c). Why the authors reported the Pvalue effect of ANOVA < 0.01 and P = 0.07? In addition, others significant differences (P < 0.05) among groups studied were not reported. Check carefully.

Res: â‘ The “P < 0.05” is the cut off level for “significant difference. The “< 0.01” indicates highly significant. We have added this information throughout the text (Statistic analysis, Tables, and Figures).

â‘¡ We have added the other significant results which have significant differences in lines 265 - 266. 

Lines 280-281: the following sentence is not correct: “In the present study, the decrease of lightness and yellowness in E120V was consistent with a higher MDA level at d 0.”.

Res: We have changed “decrease” to “increase” in line 283.

Lines 283-286: these sentences are not clear for me:. “Authors should discuss the results and how they can be interpreted in perspective of previous studies and of the working hypotheses. The findings and their implications should be discussed in the broadest context possible. Future research directions may also be highlighted.” What the authors want to say? Is it a mistake?

Res: Sorry, it is the text from the Article Template. We have deleted these words of the template after line 286.

Table 3

Title: report in the title that the data regard time 0.

Res: We have revised the table title in lines 289-290.

Figure 1 and 2: include “means ± <SD”

Res: We have added the information in line 232 and 259.

Extra revisions done by authors.

We have revised the grammar of the whole text, and highlighted them in yellow. We have add “improve”before “product” in line 65. We have add “the”before “the” in line 65. We have abbreviated “DPPHelimination ability” to “DPPH•EA”, and revised in the whole text. We have revisedthe thickness of horizontal line in tables. We have add “that in”after “than” in lines 176, 267 and 269. We have revised“get touch with” to “touch” in line 113. We have add “ (TE)”after “Trolox equivalent” in lines 157. We have add “be”after “may” in lines 184.

Once again, thank you very much for your comments and suggestions.

Best regards,

Sincerely yours,

Xin Zhang, 

College of Animal Science and Technology, Yangzhou University

Yangzhou City, Jiangsu Province, P. R. China, 225009

Reviewer 3 Report

Manuscript ID: animals-679198

 Title: Evaluation of the effects of pre-slaughter high-frequency electrical stunning current intensities on lipid oxidative stability and antioxidant capacity in the livers of Yangzhou geese (Anser cygnoides domesticus)

General comment

 The objective of the study has been to evaluate the effects of pre-slaughter electrical stunning, with different electrical intensities, on colour, lipid oxidative stability, and antioxidant capacity in meat goose livers. The experimental design is appropriate. The topic is interesting because limited research has been performed to evaluate the effects of high-frequency electrical stunning on liver of  geese. Specific comments are below reported.

Title: change "livers" with "liver".

Simple Summary

Line 23: “meet”??

Introduction

Line 67. “animal welfare” Generally animal welfare is used for living animals. Please change with other words.

M&M

Lines 85-87: Move this sentences in introduction section.

Lines 87-91: I suggest to include some information regarding the rearing condition. In addition: the animals had free access to the water?

Line 129: were the geese subjected to the fasting period?

Lines 168- 173: What does this part mean? What do the authors want to describe? It is not clear.

R&D

Line 180: change “P < 0.01” with “P < 0.05” (the differences reported among means values, shown in the table 2, are for P< 0.05_ a-b)

Lines 185-186: replace “&” with “and”

Lines 189-190: This sentence does not describe correctly the results regarding the effect of the storage time on MDA content. E60V group doesn't show statistical difference according with the letters x and y. Rewrite the sentence.

Line 194: “The results suggested that …” replace with “The results of the present study suggested that …”

Lines 207-209: the sentence “However, at 2 d postmortem, the DPPH elimination ability, the enzyme activities of CAT, GSH-PX and T-SOD in  geese livers was higher in the E60V group than in E120V groups (P < 0.05, Fig. 1B).” does not describe correctly the results from Fig. 1B (e.g. DPPH: E 30V versus E120V. CAT: Control versus E60V. T-SOD: E90V versus E120V; E60V versus E90V).

Lines 214-216: revise and complete this sentence “At 4 d postmortem, the DPPH elimination ability, the activity of T-SOD in the livers of geese was higher in the E60V groups than in the E120V group (P < 0.05, Fig. 1C), according with the results reported in Fig 1C.

Lines 262-264: All the differences showed in table 3 are for P < 0.05 (a-c). Why the authors reported the P value effect of ANOVA P<0.01 and P= 0.07? In addition, others significant differences (P < 0.05) among groups studied were not reported. Check carefully.

Lines 280-281: the following sentence is not correct: “In the present study, the decrease of lightness and yellowness in E120V was consistent with a higher MDA level at d 0.”.

Lines 283-286: these sentences are not clear for me:. “Authors should discuss the results and how they can be interpreted in perspective of previous studies and of the working hypotheses. The findings  and their implications should be discussed in the broadest context possible. Future research directions may also be highlighted.” What the authors want to say? Is it a mistake?

 Table 3

Title: report in the title that the data regard time 0.

Figure 1 and 2: include “means ± <SD”

Author Response

Dear editors and reviewers,

        Authors are very grateful for your precious time, careful and efficient review and comments for our manuscript (manuscript ID: animals-679198)! Your comments are very valuable for improving the quality of our paper! Revisions have been down according to your suggestions. Besides, the whole manuscript has been carefully checked again, and some extra revisions were made. These main changes are listed at the end of this response letter. Revisions that we did are highlighted in the text. If you have any questions, please don’t hesitate to contact me. Thank you very much!

Response (Res) to reviewer

Title: 

change “livers” with “liver”.

Res: We have changed “the livers of Yangzhou geese” with “the liver of Yangzhou goose” in the whole text, and highlighted them in yellow. 

Simple Summary:

Line 23: “meet”?

Res: We have revised “meet” to “meat” in line 21.

Introduction: 

Line 67. “animal welfare” Generally animal welfare is used for living animals. Please change with other words.

Res: We have replaced “animal welfare” with “reduce the pain of poultry” in lines 64 - 65.

M&M

Lines 85-87: Move this sentences in introduction section.

Res: We have deleted the introduction of Yangzhou geese in the material method, and make the corresponding modification in the reference of the article. 

Lines 87-91: I suggest to include some information regarding the rearing condition. In addition: the animals had free access to the water?

Res: We have added the information of rearing condition in lines 84 - 88.

Line 129: were the geese subjected to the fasting period?

Res: All the geese were weighed individually after fasting for 6 h. We have added the information in line 128.

Lines 168- 173: What does this part mean? What do the authors want to describe? It is not clear.

Res: Sorry, it is Article Template. We have deleted these words of the template after line 169.

R&D

Line 180: change “P< 0.01” with “P < 0.05” (the differences reported among means values, shown in the table 2, are for P < 0.05_ a-b).

Res: The “P < 0.05” is the cut off level for significant difference. The “< 0.01” indicates highly significant. We didn't write it clearly before. We have added this information throughout the text (Statistic analysis, Tables, and Figures). We have not changed "P < 0.01" with "P < 0.05" in line 176.

Lines 185-186: replace “&” with “and”.

Res: We have replaced “&” with “and” in line 182.

Lines 189-190: This sentence does not describe correctly the results regarding the effect of the storage time on MDA content. E60V group doesn’t show statistical difference according to with the letters x and y. Rewrite the sentence.

Res: We have revised the sentence in line187.

Line 194: “The results suggested that …” replace with “The results of the present study suggested that …”

Res: We have replaced: “The results suggested that …” with “The results of the present study suggested that …” in lines 191 - 192. 

Lines 207-209: the sentence “However, at 2 d postmortem, the DPPH elimination ability, the enzyme activities of CAT, GSH-PX and T-SOD in geese livers was higher in the E60V group than in E120V groups (P< 0.05, Fig. 1B).” does not describe the results correctly from Fig. 1B (e.g. DPPH: E 30V versus E120V. CAT: Control versus E60V. T-SOD: E90V versus E120V; E60V versus E90V).

Res: We have rewritten the sentence in lines 208 - 212.

Lines 214-216: revise and complete this sentence “At 4 d postmortem, the DPPH elimination ability, the activity of T-SOD in the livers of geese was higher in the E60V groups than in the E120V group (P< 0.05, Fig. 1C), according to with the results reported in Fig 1C.

Res: We have rewritten the sentence in lines 217 - 220.

Lines 262-264: All the differences showed in table 3 are for P< 0.05 (a-c). Why the authors reported the Pvalue effect of ANOVA < 0.01 and P = 0.07? In addition, others significant differences (P < 0.05) among groups studied were not reported. Check carefully.

Res: â‘ The “P < 0.05” is the cut off level for “significant difference. The “< 0.01” indicates highly significant. We have added this information throughout the text (Statistic analysis, Tables, and Figures).

â‘¡ We have added the other significant results which have significant differences in lines 265 - 266. 

Lines 280-281: the following sentence is not correct: “In the present study, the decrease of lightness and yellowness in E120V was consistent with a higher MDA level at d 0.”.

Res: We have changed “decrease” to “increase” in line 283.

Lines 283-286: these sentences are not clear for me:. “Authors should discuss the results and how they can be interpreted in perspective of previous studies and of the working hypotheses. The findings and their implications should be discussed in the broadest context possible. Future research directions may also be highlighted.” What the authors want to say? Is it a mistake?

Res: Sorry, it is the text from the Article Template. We have deleted these words of the template after line 286.

Table 3

Title: report in the title that the data regard time 0.

Res: We have revised the table title in lines 289-290.

Figure 1 and 2: include “means ± <SD”.

Res: We have added the information in line 232 and 259.

According to the opinions of other reviewers, there have been revised as follows.

Reviewer 1

Lines 9-16: There are some mistakes according to Instructions for the authors (Correspondence? don’t need telephone numbers?) 

Res: â‘ We have changed the “Corresponding author:” with “Correspondence” in line 15.

â‘¡We have provided the telephone number in line 15, but the format incorrect. We have correct it already in line 15.

Line 40: Use the same aberration in the whole text (DPPH•). 

Res: We have changed the “DPPH•” with “DPPH•” in the whole text, and highlighted them in yellow.

Line 44: Add liver or meat goose liver as a keyword (liver, colour…)

Res: We have added “liver” in the keywords and replaced “liver colour” with “colour” in line 40. 

Line 47: Introduction: There is nothing about storage time of goose liver in connection with oxidative stress in the Introduction, add some researches about this topic. There are no published data about liver colour. The way is this important, for customers?

Res: â‘ We think that there is no connection between storage time and oxidative stress (it refers to a state of live animals). Do you mean the connection between storage time and lipid oxidation? We have added one reference about the relationship between storage time and oxidative stress in lines 59 - 61. We have add “live” before “birds” in line 58.

â‘¡Goose liver has a large number of consumers in the market of China. Lipid oxidation products may damage the health of consumers. Liver colour is also an important factor in judging lipid oxidation and freshness of the liver.

Lines 81-82: What about storage time and its effects on…. please add. Can you write the aim in the same order than later in materials and methods, results… (colour of the liver is at the end)?

Res: â‘ The “storage time and its effects on…” has been added into the aim in lines 80 - 81.

â‘¡We have put the “colour” after antioxidant capacity in line 81.

Lines 85-87: This is sentence is a part of for Introduction, because describe Yangzhou geese and it isn’t describing the Materials and methods of the present research.

Res: This sentence was a duplicate of the first sentence in Introduction. We have deleted this sentence and its citation from the Material & Method. In addition, the reference was also deleted from the Reference section. 

Line 94: Table 1., You don’t need Total (100.00). Can you explain what ME, CP and CF mean?

Res: We have deleted the Total (100.00), and added the explanation of ME, CP and CF in line 100.

Line 129-133: What about weights of the liver (W1) – please add in the results. I can’t find these numbers (Blood loss) in the results.

Res: â‘ The W1 referred to geese weight rather than liver weight. We didn’t weigh the liver.

       â‘¡The numbers of blood loss were in table 3.

Line 134, chapter 2.4. Please add how long till analysis of lipid oxidation and antioxidant capacity samples were stored. 

Res: We measured those variables within three months after sampling. We have added the information in lines 135 - 137.

Lines 143; 145: GSH-Px or GSH-PX; use the same abbreviation in the text.

Res: We have changed the “GSH-Px” with “GSH-PX” in the whole text, and highlighted them in yellow.

Line 148: The 1, 1-Diphenyl-2-picrylhydrazyl (DPPH •), is D small letter.

Res: Yes, it should be a small letter. We have revised “1, 1-Diphenyl-2-picrylhydrazyl” to “1, 1-diphenyl-2-picrylhydrazyl” in line 149.

Line 167 – 173: I don’t understand? Materials and methods should…. This is the text from the Article Template!

Res: Yes, these sentences came from the Article Template. Sorry, it was our mistake. We have deleted these words of the template after line 169.

Line 199: .is missing at the end of the table title. What does prot mean = protein? Add in the title of the table malondialdehyde (MDA) concentration. Can you explain (in materials and methods) how you calculate MDA concentrations/mg proteins.

Res: â‘ We have added the dot at the end of the table title in line 93, 196 and 290. 

       â‘¡The “prot” means protein. We have changed the “prot” with “protein” in the whole text. 

       â‘¢The results of MDA concentrations were calculated on the basis of the protein content in liver homogenates. We have added the related information in lines 147 - 148.

Line 200: Table 2. Can you explain below the table what d 0, d 2 and d 4 mean?

Res: The d 0, d 2 and d 4, respectively means day 0, 2, and 4. We have added the information in lines 200 - 201.

Line 208: DPPH• elimination ability, above you used abbreviation (DPPH)?

Res: DPPH is short for 1, 1-diphenyl-2-picrylhydrazyl, whereas, DPPH• refers to DPPH free radical. The “DPPH• elimination ability” is abbreviated as DPPH•EA throughout the text of our revised manuscript.

Line 223, Figure 1: A, B, C up and down in the figure? There isn’t the same size and shape of all three figures and also letters. Do you really need on all figures name Antioxidative capacity?

Res: We have corrected and merged the photos A, B, C into one picture (Figure 1).

Line 251, Figure 2: A, B, C up and down in the figure? Do you really need on all figures, name Electrical stunning methods? What mean prot (protein or aberration?)

Res: â‘ We have corrected and merged these pictures into Figure 2. 

       â‘¡We have renamed all the figures in Figure 2. 

       â‘¢The “prot” means protein. We have explained it throughout the text of the manuscript.

Line 257: There is a lot of space … Livers…and ….C

Res: We have deleted the spare space in line 255.

Lines 283 -286: I don’t understand the text Authors should discuss the results and how they can be interpreted in perspective of previous studies and of the working hypotheses. The findings and their implications should be discussed in the broadest context possible. Future research directions may also be highlighted. This is the text from the Article Template!

Res: Yes, these sentences came from the Article Template. Sorry, it was our mistake. We have deleted these words after line 286.

Line 287. Dot is missing at the end of the table title. Why extra P< 0.01 in the Lightness (second row), P-value is in the first row in the table? 

Res: â‘ We have added the dot at the end of the table title in line 290. 

â‘¡The p-value is in the last column. The P-value of brightness is less than 0.01, which cannot be filled in the form accurately, so we use “p < 0.01” instead before. We have deleted the “p” of the “p < 0.01” in the second row. A similar mistake was corrected in Table 3.c.

Line 288. Dot is missing at the end of the sentence.

Res: We have added the dot at the end of the sentence in line 291.

Line 293 Conclusions; Rewrite the conclusions in order of the materials and methods and results, lipid oxidation….colour. Add what these results suggest; which combination could be applied in the ES to improve the lipid oxidative stability.

Res: â‘ We have revised “the lightness, antioxidant capacity and lipid oxidative stability” to “the antioxidant capacity, lipid oxidative stability and the lightness” in lines 298 - 299.

â‘¡We have added the suggestion of results in lines 300-301.

Lines 300-301. There are not space between the letter of the names. Please check the Article template and Instructions for the Authors in text. 

Res: Yes, we have deleted the spare spaces in lines 304 - 305.

Reviewer 2

The Authors have investigated an interesting topic and the theme has been properly described. I would like to congratulate authors for the good-quality of the article, the literature reported used to write the paper, and for the clear and appropriate structure. The manuscript is well written, presented and discussed, and understandable to a specialist readership. In general, the organization and the structure of the article are satisfactory and in agreement with the journal instructions for authors. The subject is adequate with the overall journal scope. The work shows a conscientious study in which a very exhaustive discussion of the literature available has been carried out. The introduction provides sufficient background, and the other sections include results clearly presented and analyzed exhaustively. As specific comment, the Simple Summary could be shortened. 

Res: Thank you for your suggestion. We have shortened the Simple Summary. Details are listed as followings:

â‘ We have replaced the “However, limited research has been performed to evaluate the ES effects on the liver lipid oxidative stability of the meat geese.” with “Limited research has been performed on the effect of ES on liver quality of the meat geese” in lines 19 - 20.

â‘¡We have deleted these words: “Geese were exposed to ES for 10 s with alternating current (AC) at 500 Hz in a water bath, which was combined with one of four different current intensities”. 

â‘¢ We have replaced “Stunning each goose with the current intensity at 60 V/40 mA improved the lightness, antioxidant capacity, and lipid oxidative stability in livers of meat geese during cold storage from day 0 to day four as compared with high-current ES (120 V/100 mA) based on the same high electrical frequency (500 Hz, AC).” with “Stunning each goose with the current intensity at 60 V/40 mA improved antioxidant capacity and lipid oxidative stability from day 0 to day four and the lightness at day 0 in livers of meat geese during cold storage.” in lines 22 - 24.

â‘£We have replaced “the lipid oxidative stability and antioxidant capacity” with “the antioxidant capacity and lipid oxidative stability.” in lines 25 - 26 through out the text according to the suggestion of other reviewers.

Extra revisions done by authors.

We have revised the grammar of the whole text, and highlighted them in yellow. We have add “improve”before “product” in line 65. We have add “the”before “the” in line 65. We have abbreviated “DPPHelimination ability” to “DPPH•EA”, and revised in the whole text. We have revisedthe thickness of horizontal line in tables. We have add “that in”after “than” in lines 176, 267 and 269. We have revised“get touch with” to “touch” in line 113. We have add “ (TE)”after “Trolox equivalent” in lines 157. We have add “be”after “may” in lines 184.

Once again, thank you very much for your comments and suggestions.

Best regards,

Sincerely yours,

Xin Zhang, 

College of Animal Science and Technology, Yangzhou University

Yangzhou City, Jiangsu Province, P. R. China, 225009
